# In Vitro Evaluation of Anti-Rotaviral Activity and Intestinal Toxicity of a Phytotherapeutic Prototype of *Achyrocline bogotensis* (Kunth) DC.

**DOI:** 10.3390/v14112394

**Published:** 2022-10-29

**Authors:** María-Camila Ramírez, Kelly Méndez, Alicia Castelblanco-Mora, Sandra Quijano, Juan Ulloa

**Affiliations:** 1Laboratorio de Virología, Grupo de Enfermedades Infecciosas, Departamento de Microbiología, Facultad de Ciencias, Pontificia Universidad Javeriana, Cra. 7 # 43-82, Bogotá D.C. 110231, Colombia; 2Grupo de Inmunobiología y Biología Celular, Departamento de Microbiología, Facultad de Ciencias, Pontificia Universidad Javeriana, Cra. 7 # 43-82, Bogotá D.C. 110231, Colombia

**Keywords:** *Achyrocline bogotensis* (Kunth) DC., anti-rotavirus, cytotoxicity, acute diarrheal disease

## Abstract

Viruses represent the primary etiologic agents (70–80%) of acute diarrheal disease (ADD), and rotavirus (RV) is the most relevant one. Currently, four rotavirus vaccines are available. However, these vaccines do not protect against emerging viral strains or are not available in low-income countries. To date, there are no approved drugs available against rotavirus infection. In this study, we evaluated the in vitro anti-rotaviral activity and intestinal toxicity of a phytotherapeutic prototype obtained from *Achyrocline bogotensis* (Kunth) DC. (PPAb); medicinal plant that contains compounds that inhibit the rotavirus replication cycle. Virucidal and viral yield reduction effects exerted by the PPAb were evaluated by immunocytochemistry and flow cytometry. Furthermore, the toxic impact of the PPAb was evaluated in polarized human intestinal epithelial C2BBe1 cells in terms of cytotoxicity, loss of cytoplasmic membrane asymmetry, and DNA fragmentation by MTT and fluorometry. PPAb concentrations under 0.49 mg/mL exerted significant virucidal and viral yield reduction activities, and concentrations under 16 mg/mL neither reduced cell viability, produced DNA fragmentation, nor compromised the C2BBe1cell membrane stability after 24-h incubation. Based on these results, the evaluated phytotherapeutic prototype of *Achyrocline bogotensis* might be considered as a promising alternative to treat ADD caused by rotavirus.

## 1. Introduction

Acute diarrheal disease (ADD) is an abnormally frequent discharge of semisolid or fluid stool lasting less than 14 days [1], and it is caused by infectious agents, such as bacteria, parasitic protozoa, and viruses. These organisms are commonly acquired through contaminated food and water ingestion and person-to-person contact due to poor hygiene practices. The infection can result in severe dehydration and electrolyte imbalance that produce lethargy, unconsciousness, sunken eyes, inability to drink, restlessness, and irritability. Infectious ADD can have diverse etiologic agents. Rotavirus (RV) and *Escherichia coli* are the most common agents in low-income countries, although *Shigella* spp. and *Cryptosporidium* spp. are also involved [2].

Rotavirus is a naked virus with a segmented double-stranded RNA genome that belongs to the *Reoviridae* family. The icosahedral virion is a triple-layered particle made up of several structural proteins: an outer VP7/VP4 layer, a VP6 media layer, and a VP2 inner nucleocapsid [3]. The RV genome comprises eleven genes that code for six structural (VP1 to VP4 and VP6 to VP7) and five nonstructural (NSP1 to NSP5/NSP6) proteins. NSP4 was the first viral enterotoxin described; it is a glycoprotein of 175 amino acids which plays a critical role in virion morphogenesis and pathogenicity of the particle [4].

NSP4 is secreted early by RV-infected intestinal epithelial cells and interacts with other uninfected enterocytes through receptors, such as α1β1 and α2β1 integrins. These receptors, when activated, stimulate intracellular signaling cascades that in turn increase the intracellular calcium levels, thus causing osmotic diarrhea [5], in addition to the secretory diarrhea caused by the virus on its own.

Worldwide, RV infection is responsible for the death of approximately 215,000 children under five years of age per year [6]. The conventional treatment for ADD caused by RV entails the administration of oral rehydration salts (ORS) to replenish water and electrolyte losses [2]. To date, there are no approved drugs available against rotavirus infection. Furthermore, four rotavirus vaccines (Rotarix™, Rotateq™, Rotavac™ and RotaSiil™) are currently in use [6], but some of them have shown differential efficacy between developing regions of Africa (19.6% to 64.2%) and Asia (45.5% to 64.2%), and the developed countries (>85%) [7]. These differences have been attributed to several factors, including malnutrition, host genetics, the rotavirus strain diversity [8], and the genetic reassortment of vaccine rotavirus strains with wild strains that has given rise to new RV phenotypes [9,10]. Additionally, there are reports of adverse effects associated with RV vaccination, including intestinal invagination [11,12], cough, runny nose, diarrhea, irritability, appetite loss, fever, vomiting [13], and respiratory disease [14].

An alternative approach based on medicinal plants has been suggested to treat ADD and RV infection [15,16,17]. Phytotherapeutic products can be prepared from plant extracts and are characterized by their high content of unknown active substances, wide therapeutic range (useful in chronic diseases) and lower association with side effects than synthetic drugs [18,19]. The World Health Organization has defined *herbal medicines* as “finished, labelled medicinal products that contain as active ingredients, aerial or underground parts of plants, or other plant material, or combinations thereof, whether in the crude state or as plant preparations. Plant material includes juices, gums, fatty oils, essential oils, and any other substances of this nature. Herbal medicines may contain excipients in addition to active ingredients. Medicines containing plant material combined with chemically defined active substances, including chemically defined, isolated constituents of plants, are not considered to be herbal medicines.” [20].

In 2015, our research group reported the in vitro antiviral activity of an F1 fraction phytochemically obtained from *Achyrocline bogotensis* (Kunth) DC. against rhesus rotavirus (RRV) and human astrovirus (Yuc8 strain) [21]. This traditional medicinal plant is an endemic species of Colombia, known by the vernacular names ¨vira vira¨, ¨cenizo¨, and ¨suso¨. Subsequently, the characterization of that F1 fraction by gas chromatography coupled to mass spectrometry (GC/MS) revealed a high content of flavonoids along with ethyl esters derived from fatty acids (palmitic, linoleic, linolenic, stearic, arachidic, myristic, and margaric acids), phytol compounds, and sterols (22,23-dihydro-Stigmasterol, Stigmasterol and Stigmast-4-en-3-one). Based on these results, a first batch (pilot scale) of a phytotherapeutic prototype derived from *A. bogotensis* (PPAb) was prepared industrially under Good Manufacturing Practices (GMP). This PPAb contained the dried hydroalcoholic extract of the aerial parts of the plant (leaves, stems, and flowers) and pharmaceutically acceptable levels of vehicle and excipients. The in vitro anti-rotavirus activity of non-cytotoxic concentrations (mg/mL) of PPAb was evaluated by the viral infectious foci reduction in cultures of MA104 cells [22,23]. Results from this screening showed a significant PPAb virucidal activity against RV.

These preliminary results prompted us to confirm the anti-rotaviral effect of the PPAb and evaluate its in vitro toxicity on intestinal cells. For this purpose, a second batch of the PPAb was industrially manufactured under GMP, and the major compounds of the F1 fraction were defined by GC/MS [23]. The present study describes the significant in vitro anti-rotaviral activity of PPAb and its low toxicity on human intestinal cells.

## 2. Materials and Methods

### 2.1. Cytotoxicity of PPAb for MA104 Cells, Susceptible to RV Infection

The cytotoxic effect of PPAb was evaluated on monkey kidney (MA104) cells that are susceptible to RV infection. MA104, (a kind gift from Dr. Luz Stella Rodríguez—Pontificia Universidad Javeriana, Bogotá, Colombia) cells were seeded in 96-well plates (15,000 per well) in Advanced DMEM (Invitrogen™; Grand Island, NY, USA) supplemented with 2 mM L-glutamine, antibiotics/antimycotic, and 5% fetal bovine serum (FBS; Gibco™; Grand Island, NY, USA). Cells were cultured at 37 °C in a 5% CO_2_ atmosphere for 48 h up to 90% of confluence. Then, they were exposed to fifteen different PPAb concentrations in a range of 250 mg/mL to 0.01 mg/mL (serial two-fold dilutions). All PPAb concentrations were evaluated in triplicates and experiments were repeated three times. Non-treated cells (cell control, CC) and cells exposed to 1 mM hydrogen peroxide (H_2_O_2_; cytotoxicity control) were run in parallel. Cell viability was evaluated at 48 h by the MTT (Thiazolyl Blue Tetrazolium Bromide; Sigma Aldrich Co., Ltd.; St, Louis, MO, USA) assay. The absorbance at 540 nm was measured in a MultisKan™ FC Microplate Photometer (Thermo Scientific™). The percentage of cell viability was calculated using the following formula: (% viable cells) = 100 × (sample Abs average)**/**(control Abs average).

Subsequently, the maximum non-toxic concentration (MNTC) of PPAb, i.e., the maximum PPAb concentration with no toxic effect for cells (cell viability ≥ 95%) was determined. PPAb concentrations lower than or equal to the MNTC were then used in assays designed to assess antiviral activity. All experiments were done in triplicates and repeated independently three times. Differences were analyzed with Welch’s *t*-test (*p* < 0.05) by using GraphPad Prism 6.0b software (GraphPad, San Diego, CA, USA).

The dye exclusion test (0.04% trypan blue) was also performed to observe alterations in cell morphology; data were also compared to results from the MTT assay. Photographic records were acquired using a Moticam 10+ camera (Motic^®^; Kowloon, Hong Kong) coupled to an Olympus CKX41 microscope (Olympus Instruments, Tokyo, Japan), (10× objective).

### 2.2. Anti-Rotavirus Activity of the PPAb: Virucidal Effect and Viral Yield Reduction

#### 2.2.1. Immunocytochemistry (ICC)

The virucidal activity (functional inhibition of viral infectivity) of the PPAb on infectious RV particles was assessed in MA104 cells. For this purpose, MA104 cells were cultured as previously described for 72 h up to 100% confluence, and, in addition, aliquots of an RRV suspension (previously activated with 10 µg/mL trypsin for proteolytic cleavage of the VP4 spike protein into its mature products VP8* and VP5*) were mixed with six non-cytotoxic concentrations of PPAb (1:1 ratio) at 37 °C for 2 h or 4 h. At the time of testing, MA104 cells were washed twice with sterile phosphate buffered-saline (PBS, 1×) before adding the RV-PPAb mixtures (final MOI = 0.1); final PPAb concentration range from 0.49 mg/mL to 0.01 mg/mL; cells were then incubated at 37 °C for 1 h to facilitate viral adsorption. The RV-PPAb mixtures were then removed, and the cells were washed twice with sterile PBS (1×) and incubated with culture medium without FBS for 10 h. This period was established taking into account that the replication cycle of the rotavirus-RRV strain lasts approximately 14 h, and structural proteins can be detected starting at 8 h after culture infection at an MOI of 0.1 [24]. Subsequently, the culture medium was removed, and the cells were fixed with 80% acetone in PBS for 15 min at room temperature (RT) for 15 min and washed three times with sterile PBS (1×). Then, cells were incubated with rabbit polyclonal anti-TLP antibodies (1:3000 diluted in culture medium without FBS) at RT for one hour and washed three times with sterile PBS (1×). Then, they were covered with goat anti-rabbit IgG (H + L) secondary antibody conjugated to HRP (1:3000, Invitrogen™; Rockford, IL, USA, in culture medium without FBS) at RT for 1 h and washed three times with sterile PBS (1×). Finally, a suspension of AEC (3-amino, 9-ethyl carbazole, Sigma Aldrich Co., Ltd.; St, Louis, MO, USA) and 0.02% H_2_O_2_ was added to reveal the immunoreaction, followed by two washes with distilled water. Focus-forming units (FFUs that correspond to single or clusters of infected cells) were counted using an Olympus CKX41 microscope (20× objective). Photographic records were captured with a Moticam 10+ camera (10× objective). Parallel experiments were also done with non-RV infected cells (Mock control) and non-PPAb treated RV-infected cells (infectivity control). All experiments were conducted in triplicate and repeated independently three times. Differences were analyzed with the Welch’s *t*-test (*p* < 0.05) using the GraphPad Prism 6.0b software.

Additionally, to evaluate the infectious virus yield reduction post-infection, 25,000 MA104 cells per cm^2^ were seeded in 24-well plates under the conditions previously described until 100% of confluence; then, cells were infected with trypsin-activated RV (MOI = 0.1); after 1 h of adsorption, the viral inocula were removed, and cells were washed twice with sterile PBS (1×). Immediately after, cells were exposed to non-cytotoxic PPAb concentrations in Advanced DMEM (Invitrogen™; Grand Island, NY, USA) without FBS and incubated at 37 °C in a 5% CO_2_ atmosphere for 24 h. Afterwards, viral particles were harvested by freezing at −20 °C and thawing at RT twice. The virus yield was quantified by immunocytochemistry as mentioned above for FFU counting. Non-RV infected cells (Mock control) and non-PPAb treated RV-infected cells (infectivity control) were run in parallel. All experiments were done in triplicate and repeated independently three times. Differences were analyzed with the Welch’s *t*-test (*p* < 0.05) using the GraphPad Prism 6.0b software.). The percentage of viral inhibition exerted by the PPAb was estimated as follows: (% of viral inhibition) = 100 − (UFF mean with PPAb × 100)**/**(UFF mean with non-PPAb treated RV or non-PPAb treated RV-infected cells).

#### 2.2.2. Flow Cytometry

To confirm the ICC results, similar experiments were run in duplicate and repeated independently three times to be analyzed by flow cytometry. In brief, as previously described, MA104 cells were cultured in 24-well plates, washed twice with FACSFlow™ buffer (Becton Dickinson; San Jose, CA, USA), and incubated with the RV-PPAb mixtures for 1 h to facilitate viral adsorption. Then, the RV-PPAb mixtures were removed, cells were washed twice with FACSFlow™ buffer and incubated with the culture medium without FBS for 9 h. Non-RV infected cells (Mock control) and non-PPAb treated RV-infected cells (infectivity control) were run in parallel. After incubation, the culture medium was removed, and cells were washed once with FACSFlow™ buffer, sensitized with 1 mM EDTA in FACSFlow™ buffer for 1 min, and dissociated with trypsin-EDTA at 37 °C for 9 min. Then, trypsin was neutralized with a trypsin inhibitor (Gibco^®^ Life Technologies; Grand Island, NY, USA) and mild mechanical mixing. Cells were then transferred to 1.5-mL conical tubes and centrifuged at 4500× *g* for 5 min. The cell pellet was suspended in EDTA-FACSFlow™ buffer, fixed with IntraStrain Reagent A fixative (DAKO; Glostrup, Denmark), vortexed, and incubated at RT for 15 min. Afterwards, cells were mixed with a blocking solution (MACS BSA Stock Solution/Miltenyi Biotec; Bergisch Gladbach, Germany; 1:20 in FACSFlow™ buffer) at RT for 30 min before immunostaining for cytoplasmic viral structural proteins. Thus, cells were incubated with rabbit polyclonal anti-TLP antibodies (1:3000 in IntraStrain B permeabilization reagent; DAKO) in darkness, at RT, for 1 h; washed with a blocking solution and incubated with a goat anti-rabbit IgG (H + L) highly cross-adsorbed secondary antibody conjugated with Alexa Fluor 488 (Thermo Scientific, Eugene, OR, USA, 1:3000 in IntraStrain Reagent B permeabilization reagent) in darkness, at RT, for 1 h. After two washes, cells were centrifuged, suspended in FACSFlow™ buffer, and analyzed in a FACS Aria II flow cytometer (BD Biosciences). Cytometer compensation was performed using CompBeads™ (BD Biosciences) to set the fluorescence signals and voltages for optimal discrimination between positive and negative signals. The daily cytometer calibration was done with Cytometer Setup & Tracking Beads (CST, BD Biosciences) following the manufacturer’s recommendations. Data analyses were run in the Infinicyt™ software (Cytognos SL), and differences were analyzed by the Mann–Whitney test (*p* < 0.05) using the GraphPad Prism 6.0b software.

Likewise, to assess the infectious virus yield reduction post-infection, MA104 cells were cultured in 24-well plates and infected with trypsin-activated RV (MOI = 0.1). After 1 h-incubation to facilitate viral adsorption, the inocula were removed, and the cells were washed with Advanced DMEM without FBS. Cells were then treated with the non-toxic PPAb concentrations prepared in culture medium without FBS and incubated at 37 °C and 5% CO_2_ for 9 h. Non-RV infected cells (Mock control) and non-PPAb treated RV-infected cells (infectivity control) were carried out in parallel. The assays were performed four times, independently, in duplicate. The staining for RV structural proteins was performed as mentioned previously, and cells were acquired on a FACS Aria II flow cytometer (BD Biosciences), as mentioned above. The percentage of viral inhibition exerted by the PPAb was estimated in the same manner as indicated above.

### 2.3. PPAB Toxicity for Human Intestinal Cells: C2BBe1 Cells

#### 2.3.1. Cytotoxicity Assay

Human intestinal C2BBe1 cells (ATCC^®^ CRL-2102™) were cultured for 14 days to achieve polarization. Then, cells were incubated with fifteen PPAb concentrations in a range of 250 mg/mL to 0.01 mg/mL (serial two-fold dilutions, in fresh culture medium without FBS) at 37 °C and 5% CO_2_ for 48 h. Non-PPAb treated cells (cell control, CC) and cells exposed to 50 mM H_2_O_2_ (positive control) were arranged in parallel. Experiments were performed in triplicate. Subsequently, they were analyzed by the MTT assay as previously described, and the MNTC of the PPAb was calculated.

#### 2.3.2. Evaluation of Death Markers Exerted by the PPAb on C2BBe1 Cells: Cytoplasmic Membrane Asymmetry and DNA Fragmentation

Polarized human intestinal C2BBe1 cells were cultured in 96-well flat clear bottom black polystyrene TC-treated Microplates (Costar^®^, Kennebunk, ME, USA) and exposed to five PPAb (250, 62.5, 15.62, 3.9, and 0.97 mg/mL) at 37 °C and 5% CO_2_ for 24 h. Non-PPAb treated cells (cell control, CC) were run in parallel. After 24 h, plasma membrane asymmetry (phosphatidylserine translocation to the outer leaflet of the cell membrane) and DNA fragmentation were evaluated with Annexin V and propidium iodide, respectively, with the ApoDETECT™ Annexin V-FITC Kit (Life technologies™, Frederick, MD, USA). The fluorescence signals were measured in a FLUOStar^®^ Omega Plate Reader (BMG LABTECH, Allmendgrün, Ortenberg, Germany) as relative fluorescence units (RFU). Experiments were conducted in triplicate and repeated independently three times. Differences were analyzed by the unpaired Welch’s *t*-test (*p* < 0.05) using the GraphPad Prism 6.0b software.

## 3. Results

### 3.1. Cytotoxic Effect of the PPAb on MA104 Cells

The cytotoxic effect of the PPAb (250 mg/mL to 0.01 mg/mL) on confluent MA104 cell cultures was evaluated after 48 h incubation by MTT and trypan blue dye exclusion assays. The cytotoxicity control with H_2_O_2_ showed 100% cell death. The MNTC of PPAb calculated from data of the MTT assay was 3.91 mg/mL (Figure 1A). Moreover, in cultures exposed to PPAb concentrations under 15.6 mg/mL, only a few cells internalized the trypan blue dye, and most cells preserved their morphology (Figure 1B).

### 3.2. Anti-Rotavirus Activity of the PPAb

The anti-rotavirus activity of PPAb was evaluated using two strategies: virucidal effect (functional inhibition of viral infectivity) and the reduction of viral yield post-infection.

The virucidal effect of the PPAb on infectious RV particles was assessed by ICC upon incubating MA104 cells with RRV-PPAb mixtures (preincubated for 2 h or 4 h) for 10 h. All doses evaluated (0.49 mg/mL to 0.015 mg/mL) significantly reduced the expression of RV proteins in average ranges from 51% to 71% (2 h) and 54% to 72% (4 h) (Figure 2).

It is important to note that the lowest PPAb concentration (0.015 mg/mL) exhibited the greatest inhibitory effect on viral infectivity. Moreover, PPAb concentrations from 0.49 mg/mL to 0.015 mg/mL significantly reduced the viral yield (FFU) (55% and 71%) in cultures of RV-infected MA104 (Figure 3).

Additionally, the virucidal activity of the PPAb was evaluated by flow cytometry in MA104 cells incubated with RV-PPAb mixtures (previously incubated for 2 h). The mean fluorescence intensity (MFI) of cytoplasmic RV proteins was reduced, mainly at the two highest concentrations (0.49 mg/mL and 0.24 mg/mL) of PPAb with a percentage of viral inhibition of 37% and 36%, respectively (Figure 4A).

Likewise, the rotavirus yield was reduced in an average range of 28% to 46% when the concentrations of PPAb previously mentioned were added to RV-infected MA104 cells (Figure 4B).

### 3.3. Intestinal Toxicity of the PPAb on C2BBe1 Cells: Cytotoxicity, Cytoplasmic Membrane Asymmetry and DNA Fragmentation

The toxicity of the PPAb (250 mg/mL to 0.01 mg/mL) was evaluated on polarized human intestinal C2BBe1 cells in terms of mitochondrial metabolism, membrane asymmetry, and DNA fragmentation. The MNTC calculated for the PPAb was 62.5 mg/mL by the MTT assay. However, concentrations of 62.5 mg/mL and 31.25 mg/mL increased the metabolic activity significantly at 48 h (Figure 5). Moreover, PPAb concentrations under 15.6 mg/mL did not compromise the membrane asymmetry or generate DNA fragmentation at 24 h (Figure 6).

## 4. Discussion

In 2015, our research group reported the anti-rotavirus activity of a fraction (F1 Fraction) isolated from *A. bogotensis*. This fraction contained steroids, sterols, terpenes, phenols, sesquiterpene lactones, and flavonoids [21]. In that study, mixtures of the F1 fraction (15.6 μg/mL to 0.98 μg/mL) and RV (coincubated by 1 h, 2 h, or 4 h) decreased the virus infectivity for MA104 cells (MOI = 5) by up to 42%. Based on those findings, we decided to formulate and manufacture a phytotherapeutic prototype (PTP) suitable for oral administration that contained the anti-rotavirus F1 fraction. This PTP was prepared from the total extract of *A. bogotensis* (PPAb) and was evaluated to determine its antiviral activity in vitro. As the F1 fraction had shown to represent less than 20% of the total extract of *A. bogotensis* [21], the PPAb was evaluated in higher concentrations (mg/mL) to ensure the availability of the active F1 components for the in vitro experiments.

The first batch (pilot scale) of the PPAb (manufactured under GMP) exerted significant virucidal activity against RV. This activity was evidenced by the reduced count of FFU in RV-infected MA104 cells determined by ICC; the percentage of inhibition of RV infectivity varied between 25% to 53% (2 h) and 33% to 61% (4 h) [22]. Therefore, to verify the reproducibility of the anti-rotavirus activity of the PPAb, a second batch was manufactured under similar industrial conditions, and its anti-rotavirus activity was evaluated in terms of virucidal effect and reduction of infectious virus yield post-infection, as previously described [21].

The present study showed that this second batch of the PPAb also exhibited a significant virucidal effect against RV, with infectivity reductions that ranged from 51% to 71% (2 h) and 54% to 72% (4 h). Besides, the PPAb reduced the RV yield post-infection (Figure 2, Figure 3 and Figure 4A,B). This second batch of the PPAb showed higher antiviral activity than the first one; the difference between them reached 26–18% for 2 h and 21–11% for 4 h. These differences could be attributed to a higher concentration of active antiviral components in the plant material used to obtain the second PTP batch. It is currently recognized that a poor quality control in the manufacture of herbal products diminishes their efficacy when compared to conventional chemical products [25,26]. This disadvantage is due to the scarcity of standardized crops of medicinal species. To guarantee the supply of plant raw materials and the availability and traceability of the antiviral active components of *A. bogotensis*, we are currently working on the implementation of the agrotechnology of the plant and the standardization of extracts.

The present results showed that the lowest concentration of PPAb evaluated (0.015 mg/mL or 15 µg/mL) exhibited the highest virucidal activity as evidenced by ICC. In our previous study [21], the flow cytometry analysis had shown that 15.6 µg/mL of the antiviral F1 fraction of *A. bogotensis* exerted virucidal activity, although the most active concentration was 3.91 µg/mL. These data suggest that some *A. bogotensis* components (e.g., pro-oxidant components) could inhibit the antiviral activity (synergistic activity). Therefore, we consider that it is necessary to undertake further studies to determine the PPAb concentrations that do not produce oxidative stress in rotavirus-infected cells in order to improve the antiviral effect of the prototype. Moreover, low concentrations of the PPAb might improve the selectivity index and the industrial production yield.

The compounds of *A. bogotensis* responsible for the anti-rotavirus activity have not been studied yet. Notwithstanding, flavonoids are good candidates because they can reduce cell infection by several viruses, RNA and DNA, through different strategies. These include inhibition of virus adsorption, entry and binding; and blocking of reverse transcriptase, integrase, protease, DNA and RNA polymerases (human immunodeficiency, herpes simplex, respiratory syncytial, varicella-zoster, and influenza viruses [27]. Flavonoids may also act through the induction of antiviral cytokines and interferon-stimulated genes (norovirus) [28] and reduction of early and late viral protein levels (cytomegalovirus) and viral DNA synthesis [29]. Likewise, some studies have demonstrated the anti-rotavirus activity of some flavonoids [30,31,32]. For instance, diosmin and hesperidin can inhibit the spread of RV and protect cells against viral infection [33]. Additionally, flavonoids can inhibit cellular α-glucosidase and induce misfolding of rotavirus VP7 and NSP4 glycoproteins thus interfering with RV replication [30]. These observations could explain the viral neutralization or virucidal activity of the *A. bogotensis* compounds. The F1 fraction obtained from *Achyrocline bogotensis* contained flavonoids [21,23], and the 5,7 dihydroxy-3,6,8 trimethoxyflavone and 3,5-dihydroxy-6,7,8-trimethoxy flavone were isolated from it [34,35].

In the present work, the PPAb showed significant direct virucidal and post-infection effects against RV, as evidenced by the reduced expression of the cytoplasmic viral structural proteins and reduced yield of infectious viral particles post-infection, respectively. Based on these results, it is possible to hypothesize that: (i) some *A. bogotensis* components could adhere to the virus surface and thus decrease the entry of infectious particles into the host susceptible cells; and (ii) some *A. bogotensis* components could be internalized by the host susceptible cells and leads to a reduced expression of viral proteins after infection. This effect might be related to the antioxidant activity of the flavonoids that could act by reducing the oxidative stress produced by RV [36,37]. Flavonoids, the major components of the F1 fraction isolated from *A. bogotensis,* could be involved in this inhibitory effect. As the FDA recommends identifying the mechanism of action of manufactured antiviral products [38], additional experiments should be performed to determine the mechanisms responsible for the anti-rotavirus activity of *A. bogotensis*. Currently, we are synthesizing the two main flavonoids found in the plant antiviral fraction for further analyses.

Several plant species have been reported of having anti-rotavirus activity. In 2001, Takahashi et al. showed that compounds of aqueous extracts derived from *Stevia rabaudiana* could inhibit the adhesion of RV to MA104 cells in a mechanism dependent on the structural protein VP7, which binds to integrin-type receptors [16]. Later, in 2010, the combination of bovine and porcine RV particles with several extracts derived from *Alpinia katsumadai* reduced their infectivity for MA104 cells [39]. Extracts of *Sophora flavescens* and a stevioside from *Stevia rebaudiana* Bertoni also exhibited in vitro and in vivo anti-rotavirus activity [30].

*A. bogotensis* has been previously shown to have low cellular toxicity [21,23,40]. However, competent international authorities, such as the Organization for Economic Co-operation and Development (OECD), European Union (EU) Authorities, the International Conference on Harmonization (ICH), and the World Health Organization (WHO), have continuously recommended to adopt international guidelines for the evaluation of the toxicity of plant-based products [41]. The procedures include several in vitro tests (e.g., cytotoxicity, genotoxicity, gene mutation, micronucleus, and apoptosis); and in vivo tests (e.g., mammalian bone marrow chromosome aberration and acute/subacute/chronic/subchronic reproductive and developmental toxicity). Under these critical requirements, the present work assessed the potential toxicity of the PPAb following the international guidelines from ISO 10993-5:2009 and the Interagency Coordinating Committee on the Validation of Alternative Methods (ICCVAM) [41]. Specifically, cytotoxicity, induction of apoptosis, and necrosis were tested here using the polarized human intestinal cell line, C2BBe1. Despite its tumour origin, this cell line has been used for toxicological screening [42], nanotoxicity testing [43], and plant extract toxicity assays [44]. Data from the present study showed that PPAb concentrations under 15.6 mg/mL did not decrease cell viability or produce deleterious effects on plasma membrane stability and DNA integrity of the polarized C2BBe1 cells. Although the PPAb concentrations here evaluated (mg/mL) are considered high for in vitro cell models, this work showed low in vitro toxicity for the human intestinal C2BBe1 cells (MNTC = 62.5 mg/mL) and the green monkey kidney MA104 cells (MNTC = 3.91 mg/mL). However, the finding that 62.5 mg/mL and 31.25 mg/mL of PPAb induced a significant increase in the mitochondrial activity of C2BBe1 cells at 48 h suggests that higher concentrations of the PPAb can exert a pro-oxidant intracellular environment that could eventually inhibit the antiviral effect.

Although the published investigations on the antiviral properties of plant extracts are abundant, the studies designed to verify the biological activity and toxicity of phytotherapeutic prototypes are limited. In this study, we found that the CC50 of PPAb was 553.5 mg/mL and the IC50 was 0.26 mg/mL. Thus, the selectivity index (SI) that corresponds to the CC50/IC50 ratio was 2129. The significant percentage reduction in the rotavirus infectivity observed here suggests that the in vivo administration of repeated doses of the PPAb might reduce the RV infection and, therefore, might also reduce the diarrheal episodes and the time course of the disease. Currently, our group is involved in the controlled cropping of *A. bogotensis* and the production of standardized extracts to evaluate their efficacy and toxicity in vivo.

Results from this study demonstrated for the first time a significant anti-rotavirus activity of an industrially manufactured phytotherapeutic prototype obtained from *Achyrocline bogotensis* and presented new evidence of its low toxicity for human intestinal cells in vitro.

## 5. Conclusions

Scientific evidence should support the use of pharmaceutical derivates from traditionally used medicinal plants. Data from the present research support the potential use of a phytotherapeutic prototype obtained from *A. bogotensis* to treat acute diarrhoeal disease caused by rotavirus and associated symptoms.

## Figures and Tables

**Figure 1 viruses-14-02394-f001:**
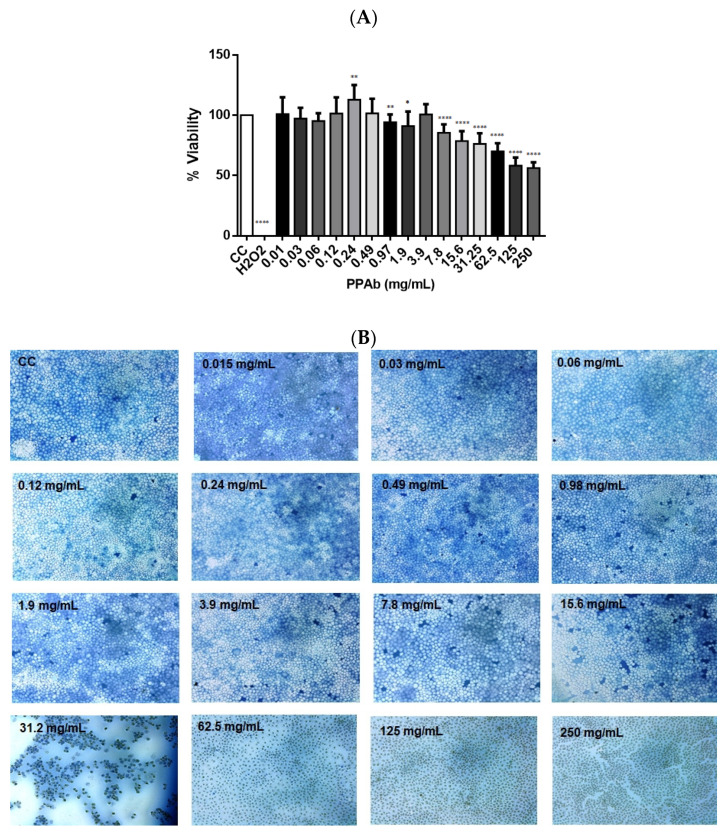
Cytotoxic effect of the PPAb on MA104 cells. MA104 cells were exposed to different concentrations of PPAb for 48 h. (**A**) Cell viability was assessed by MTT assay. CC: cell control (non-PTP treated cells); H_2_O_2_: cytotoxicity control. Bars represent mean ± SD. * *p* < 0.05, ** *p* < 0.008, **** *p* < 0.0001; unpaired Welch’s *t*-test for comparisons with the CC control; (**B**) Cell viability was evaluated by the trypan blue dye exclusion test. Representative photographs (100×). Both tests were done in triplicates and repeated independently three times.

**Figure 2 viruses-14-02394-f002:**
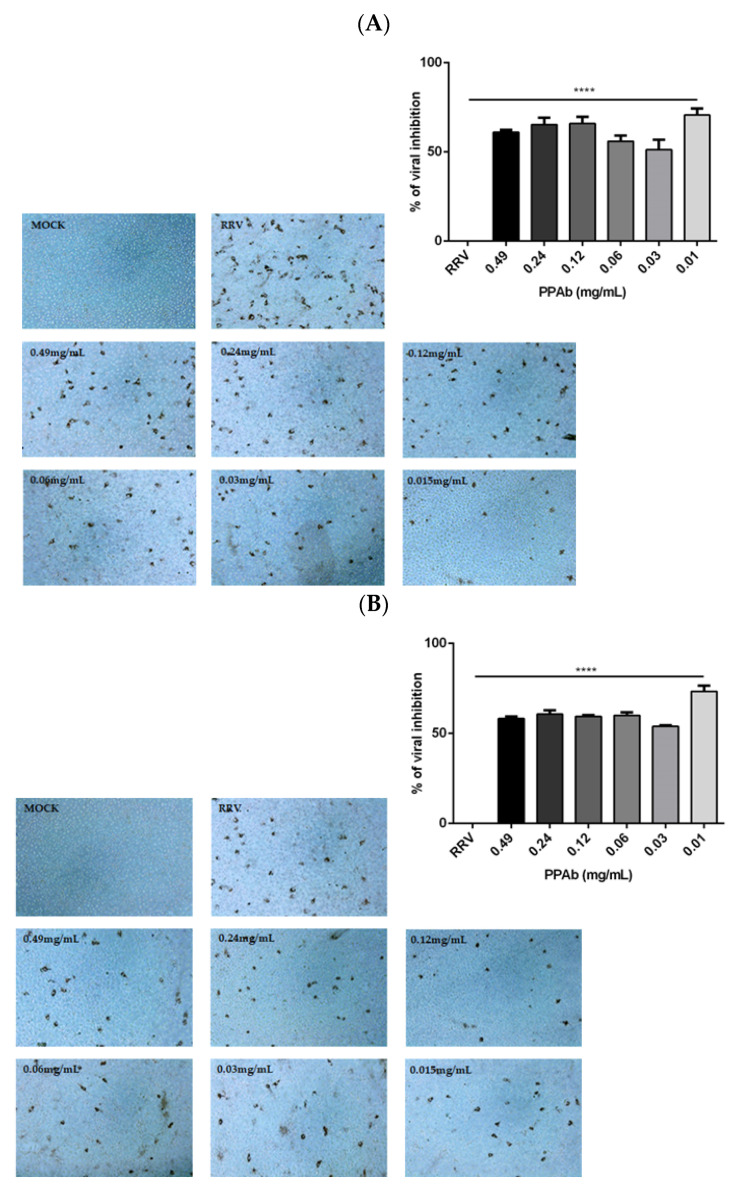
Virucidal activity of the PPAb against rotavirus. Different concentrations of PPAb were incubated with infectious RV (RRV/final MOI = 0.1) particles for 2 h (**A**) or 4 h (**B**). PPAb -RV mixtures were added to MA104 cells for 1 h at 37 °C and 5% CO_2_ for 1 h. The inocula were then removed and replaced by fresh culture medium without FBS, and incubated for 10 h. Infectious focus forming units (FFUs) were quantified by staining of intracellular RV capsid proteins using rabbit polyclonal anti-TLP antibodies (primary antibody), goat anti-rabbit IgG (H + L) antibody conjugated to HRP (secondary antibody), and AEC (3-amino, 9-ethyl carbazole) with 0.02% H_2_O_2_. MOCK: non-RRV infected cells; RRV: RV-infected cells. Bars represent mean ± SD. **** *p* < 0.0001 for comparisons with the control of RRV-infected cells; unpaired Welch’s *t*-test. All experiments were done in triplicates and repeated independently three times. Representative photographs of each condition (100×).

**Figure 3 viruses-14-02394-f003:**
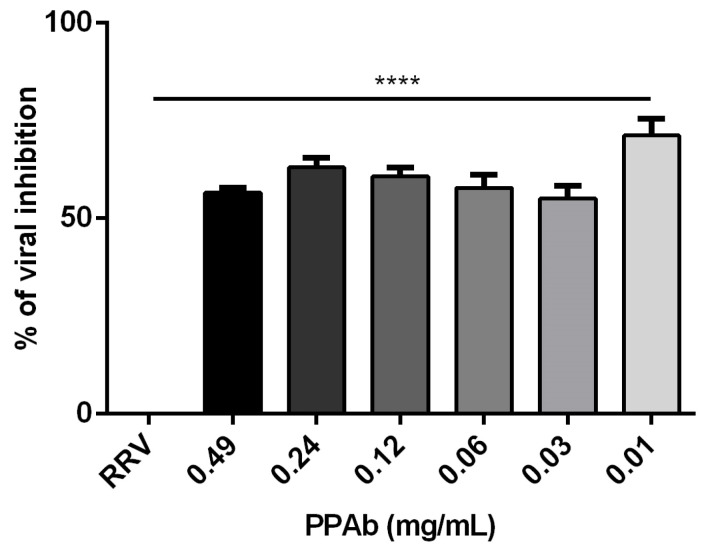
Reduction of infectious rotavirus yield post-infection exerted by the PPAb. MA104 cells were incubated with RV (RRV, MOI = 0.1) for 1 h to facilitate viral adsorption. The viral inocula were removed and replaced with culture medium without FBS and different concentrations of the PPAb. Cells were incubated 24 h to verify viral replication. Then, viral particles were harvested, and infectious FFU units (FFU) were quantified by ICC as described in Figure 2. RRV: RV-infected cells. All experiments were done in triplicates and repeated independently three times. Bars represent mean ± SD., **** *p* < 0.0001; unpaired Welch’s *t*-test for comparisons with the control of RRV-infected cells.

**Figure 4 viruses-14-02394-f004:**
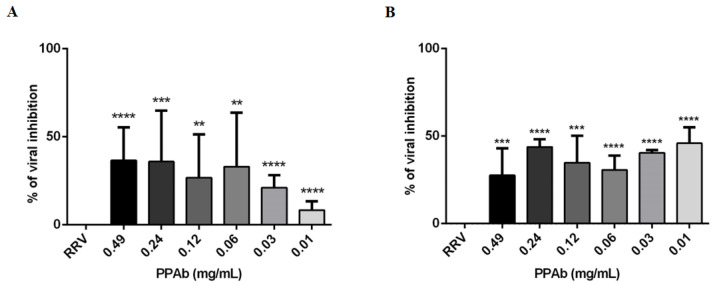
Anti-rotavirus activity of the PPAb analyzed by flow cytometry. (**A**) Virucidal effect: Different concentrations of PPAb were incubated with infectious RRV (RRV) particles at 37 °C for 2 hrs. PPAb-RV mixtures were added to MA104 cells (final MOI = 0.1) for at 37 °C and 5% CO_2_ for 1 h. Then, the inocula were removed and replaced by fresh culture medium without FBS and incubated for 9 h. (**B**) Post- infection anti-rotavirus activity: MA104 cells were incubated with RRV (MOI = 0.1) for 1 h to facilitate viral adsorption. The viral inocula were then removed and replaced with culture medium without FBS and different concentrations of the PPAb. Cells were incubated at 37 °C and 5% CO_2_ for 9 h to verify viral replication. For both assays cells were detached with trypsin-EDTA, stained for cytoplasmic RV proteins using rabbit polyclonal anti-TLP antibodies (primary antibody) and a goat anti-rabbit IgG (H + L) antibody conjugated with Alexa Fluor 488 (secondary antibody), and acquired on a flow cytometer; 10,000 events were analyzed for each condition. MOCK: Non-RRV infected cells; RRV: RRV-infected cells. Experiments were done in duplicates and repeated independently four times. Bars represent mean ± SD. ** *p* < 0.01, *** *p* <0.001, **** *p* < 0.0001; Mann-Whitney test for comparisons with the control of RRV-infected cells.

**Figure 5 viruses-14-02394-f005:**
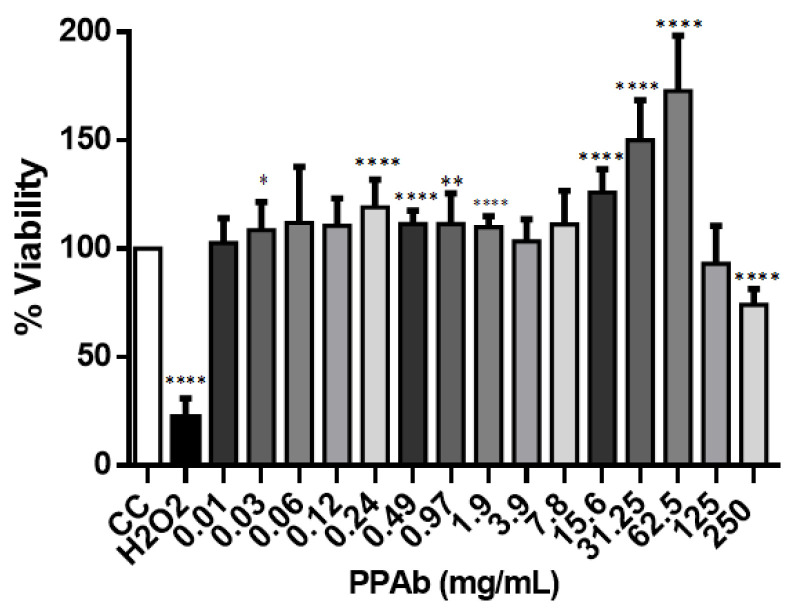
Cytotoxicity of the PPAb for human intestinal C2BBe1 cells. Polarized C2BBe1 cells were exposed to different concentrations of the PPAb at 37 °C and 5% CO_2_ for 24 h. Cell viability was assessed by MTT assay. CC: cell control (non- PPAb treated cells); H_2_O_2_: cytotoxicity control. Bars represent mean ± SD. * *p* <0.05, ** *p* <0.01, **** *p* <0.0001; unpaired Welch’s *t*-test for comparisons with the CC control.

**Figure 6 viruses-14-02394-f006:**
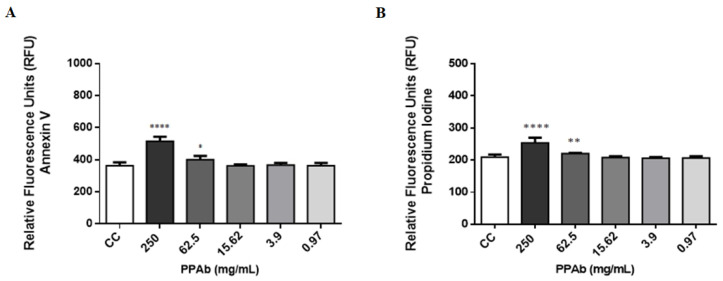
Cytoplasmic membrane asymmetry and DNA fragmentation exerted by the PPAb on human intestinal C2BBe1 cells. Polarized C2BBe1 cells were exposed to different concentrations of PPAb at 37 °C and 5% CO_2_ for 24 h. Cells were stained with the ApoDETECT™ kit (Invitrogen™) and analyzed by fluorometry. (**A**) Phosphatidylserine in the outer layer of cell membrane (Annexin V). (**B**) DNA fragmentation (Propidium Iodide). CC: cell control (non-PPAb treated cells). Experiments were done in triplicates and repeated independently three times. Bars represent mean ± SD. * *p* < 0.05, ** *p* < 0.01, **** *p* < 0.0001; unpaired Welch’s *t*-test for comparisons with the CC control.

## Data Availability

Not applicable.

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
