# Peer review of "In Vitro Evaluation of Anti-Rotaviral Activity and Intestinal Toxicity of a Phytotherapeutic Prototype of Achyrocline bogotensis (Kunth) DC."

_viruses, 2022, doi:10.3390/v14112394_

Round 1

Reviewer 1 Report

In this manuscript, virucidal effects exerted by the PPAb was evaluated in MA104 cells, the toxic impact of the PPAb was evaluated in polarised human intestinal epithelial C2BBe1 cells in terms of cytotoxicity, loss of cytoplasmic membrane asymmetry, and DNA fragmentation by MTT and fluorometry.

Major concerns:

1. To evaluate the virucidal effects by the PPAb, TCID50 or plaque assays should be used. The authors reported the virucidal efficacy by comparing the infectivity of the phytotherapeutic prototype (PTP) treated group with the control group (RV treated and set the infection rate as 100%), this manner was unsure to show the antiviral activity.

2. PPAb at a concentration of 15.6 mg/mL did not affect the membrane asymmetry or generate DNA fragmentation, these results did not give any information about the antiviral mechanism of PTP.

3. The work was rough and did not give more information than the previous work (Téllez et al. BMC Complementary and Alternative Medicine, 2015, 15:428).

Author Response

Major concerns:

  1. To evaluate the virucidal effects by the PPAb, TCID50 or plaque assays should be used. The authors reported the virucidal efficacy by comparing the infectivity of the phytotherapeutic prototype (PTP) treated group with the control group (RV treated and set the infection rate as 100%), this manner was unsure to show the antiviral activity.

Answer: We appreciate the reviewer's comment. The assays were performed by foci-forming units counting by immunocytochemistry (ICC), which is equivalent to plaque assays. Additionally, flow cytometry analyses were performed due to their high sensitivity in order to compare their results with those obtained by ICC. The results from the antiviral activity assays were reviewed, and the graphs were modified to show the percentage of viral inhibition. The formula to determine this percentage of inhibition was added (Lines 167 - 170). Figure legends were also modified.

  1. PPAb at a concentration of 15.6 mg/mL did not affect the membrane asymmetry or generate DNA fragmentation, these results did not give any information about the antiviral mechanism of PTP.

Answer: This test was part of the cytotoxicity evaluation of PPAb on human intestinal cells; it was not part of the evaluation for the antiviral activity or any possible antiviral mechanism of the PPAb.

  1. The work was rough and did not give more information than the previous work (Téllez et al. BMC Complementary and Alternative Medicine, 2015, 15:428).

Answer: We appreciate the comment of the reviewer. In the previous work by Tellez et al., three extracts and three fractions were obtained from A. bogotensis by phytochemical fractionation. The antiviral activity of those fractions was evaluated, and the antiviral fraction was partially characterized. The most relevant result of that work was the discovery of an antiviral fraction enriched in flavonoids. The present work evaluated the antiviral activity of an industrially prepared phytotherapeutic product that contained an optimized total extract (not fractionated) from A. bogotensis and excipients. As shown in the manuscript, this new product also reduces rotavirus infection in vitro and shows low cytotoxicity (MTT/Cell membrane stability/DNA fragmentation) on human intestinal cells. Thus, the present study describes new findings about the anti-rotavirus activity of an industrial prototype obtained from a natural product and its cytotoxicity in an in vitro model of human intestinal cells. In the Discussion section, the following sentence is included: Lines 460 – 462 “Although the published investigations on the antiviral properties of plant extracts are abundant, the studies designed to verify the biological activity and toxicity of phytotherapeutic prototypes are limited”. Below a picture of the product analyzed can be observed.

Reviewer 2 Report

                         Comments for the authors 

 The authors evaluated the in vitro anti-rotaviral activity and intestinal toxicity of a phytotherapeutic prototype obtained from Achyrocline bogotensis (Kunth) DC. (PPAb). Virucidal and viral yield reduction exerted by the PPAb was evaluated by immunocytochemistry and flow cytometry. Furthermore, the toxic impact of the PPAb was evaluated in polarised human intestinal epithelial C2BBe1 cells in terms of cytotoxicity, loss of cytoplasmic membrane asymmetry, and DNA fragmentation by MTT and fluorometry. 0.49 mg/mL PPAb exerted significant virucidal and viral yield reduction activities, and concentrations under 16 mg/mL reduced neither cell viability, produce DNA fragmentation, nor compromise the C2BBe1cell membrane stability after 24-h incubation. The authors finally make a conclusion that PPAb may be a promising candidate against rotavirus infection. 

Major concerns:

1.       The assay applied in this study could not verify that PPAb exerts virucidal effect. You have to design exp. to demonstrate that the virus is inactivated by PPAb before it is added into the cells. The effective dose PPAb(high concentration, A conc.) is incubated with virus, given PPAb( A conc.) can inactivate the virus directly via incubation at room temperature or 37℃, when the incubated mixture( PPAb/virus) is diluted to ineffective dose (low concentration, B conc.), the virus has no infectious. To set a comparable control, PPAb (A conc.) is directly diluted to the B conc. without incubation with virus, virus in these two concentrations is alive. After the diluted incubated mixture (virus and PPAb) is added into the permissive cells, the inactivated virus can not propagate, whereas, after mixing of the same amount of virus and the diluted PPAb (B conc.), the mixture is added into the cells, the viurs can propagate, we can conclude that PPAb has virucidal effect.

2.       Figure 1A, 0.97mg/ml and 1.9mg/ml PPAb induced significant decrease of cell viability, but the authors thought that the maximum non-toxic concentration (MNTC) of ppAb was 3.91mg/ml.

3.       About the evaluation of the antiviral candidates, TC50, IC50, seletivity index (SI) should be calculated.

4.       It should be better to choose different doses of virus inoculation with the same  concentraion of PPAb to evaluate its anti-rotaviral effect.

5.       Figure2, A and B, there is not obvious dose dependent manner of PPAb against rotavirus. The dose gradients may be required to be rescheduled.

6.       The authors mentioned that PPAb can reduce infectious virus particles post-infection, the underlying mechanism may be the same, within 24h, virus entered the cells may have produced and released progeny virus in the supernatants, which could be inactivated by the ppAb.

7.       To reveal that the life cylce of virus PPAb targets, some tools such as pp, replicon, time of addition exp.and fusion system should be applied.

8.       Pharmacokinetic studies and tissue distribution of PPAb, antivial effect evulation performed in animals may be better than simple in vitro toxicity on human intestinal cells.

Minor concerns:

1.       line 231 “conducted in triplicate were analised and statistical significance (p < 0.05) was determined by”  “analised” should be “analyzed”.

2.       For the figure legends, different graphs were marked as capitalized A/B, while in the results part, they are written in lower-case letters(a/b)

Author Response

Major concerns:

  1. The assay applied in this study could not verify that PPAb exerts virucidal effect. You have to design exp. to demonstrate that the virus is inactivated by PPAb before it is added into the cells. The effective dose PPAb(high concentration, A conc.) is incubated with virus, given PPAb( A conc.) can inactivate the virus directly via incubation at room temperature or 37℃, when the incubated mixture( PPAb/virus) is diluted to ineffective dose (low concentration, B conc.), the virus has no infectious. To set a comparable control, PPAb (A conc.) is directly diluted to the B conc. without incubation with virus, virus in these two concentrations is alive. After the diluted incubated mixture (virus and PPAb) is added into the permissive cells, the inactivated virus can not propagate, whereas, after mixing of the same amount of virus and the diluted PPAb (B conc.), the mixture is added into the cells, the viurs can propagate, we can conclude that PPAb has virucidal effect.

Answer: We sincerely appreciate the reviewer's comment and the valuable suggestion he has made. Unfortunately, at this time, we cannot do additional testing on the PPAb because it was manufactured in 2016. We recently obtained a grant to grow the plant sustainably and to produce a new batch of the PPAb under GMP. We will consider this valuable suggestion in the future. However, we think that the data obtained do reveal a direct activity of the PPAb in the reduction of RV infection in vitro, as evidenced by the infectious foci evaluated by ICC and the viral infection tested by flow cytometry. Additionally, in the Materials and Methods and Results sections, we explain that the virucidal activity evaluated relates to the functional inhibition of viral infectivity (Lines 127 and 259). The term "virucidal" refers not only to the disruption of infectious viral particles, but also to those activities that interact with viral infectivity and functionally inhibit (neutralize) it without apparent morphological alterations (Bio Protoc. 2018 May 20;8(10):e2855. doi: 10.21769/BioProtoc.2855. PMID: 34285972; PMCID: PMC8275317.). We have tried to obtain evidence of the interactions and effects of the PPAb compounds on RV particles by TEM, and it has not been possible due to the complexity of the product; the quality of the images obtained has not been appropriate.

  1. Figure 1A, 0.97mg/ml and 1.9mg/ml PPAb induced significant decrease of cell viability, but the authors thought that the maximum non-toxic concentration (MNTC) of ppAb was 3.91mg/ml.

Answer: We appreciate this reviewer's comment. Indeed, we report that 3.91 mg/mL corresponds to the MNTC according to our definition in lines 115 and 116: “…the maximum PPAb concentration with no toxic effect for cells (cell viability ≥ 95%) was determined.”. Therefore, the cell viability observed with 0.97, 1.9 and 3.91 mg/mL of PPAb was ≥ 95%; however, the standard deviation generated the significant differences commented.

  1. About the evaluation of the antiviral candidates, TC50, IC50, seletivity index (SI) should be calculated.

Answer: The parameters suggested are included at the end of the discussion section, Lines 460 – 464.

  1. It should be better to choose different doses of virus inoculation with the same concentration of PPAb to evaluate its anti-rotaviral effect.

Answer: We appreciate the reviewer's comment. Indeed, the suggestion is useful as an experimental control for the virucidal effect. In our experimental design, we used a single MOI of 0.1 so that the concentrated and diluted active PPAb compounds were more likely to interact with the virus because they represented less than 20% of the total extract. Furthermore, it is important to note that the MOI we used corresponds to 1x104 FFU, so the inhibition of rotavirus infection we found is significant.

  1. Figure2, A and B, there is not obvious dose dependent manner of PPAb against rotavirus.The dose gradients may be required to be rescheduled.

       Answer: All experiments were repeated independently with the corresponding replicates, and the data obtained were consistent. Moreover, in our previous study (Tellez et al. BMC Complementary and Alternative Medicine, 2015, 15:428), we observed similar inhibition of infection, i.e., although it was not dose-dependent, it showed statistically significant differences as a function of time. Furthermore, in the present study, the lowest concentration of PPAb showed the highest virucidal activity after 2h and 4h of coincubation with RV, as evidenced by ICC.  This finding is discussed in the manuscript in lines 387 to 396. We improved the analysis of this issue, given the importance of considering that phytotherapeutic products contain a mixture of components that act synergistically (J. Nat. Prod. 1992, 55, 12; Viruses 2018, 10(9)).

  1. The authors mentioned that PPAb can reduce infectious virus particles post-infection, the underlying mechanism may be the same, within 24h, virus entered the cells may have produced and released progeny virus in the supernatants, which could be inactivated by the PPAb.

Answer: We agree with the reviewer. As mentioned in the Discussion, we consider flavonoids may be involved in this behavior because they have been massively reported. We are currently synthesizing the two main flavonoids isolated from the active fraction to analyze the possible mechanisms involved in direct (virucidal) and post-infection inhibition. In the manuscript, we add some antiviral mechanisms reported in the literature including viral adsorption and suggested that antiviral activity may be exerted by flavonoids (Lines 400 – 402).

  1. To reveal that the life cylce of virus PPAb targets, some tools such as pp, replicon, time of addition exp. and fusion system should be applied.

Answer: We appreciate the reviewer's suggestion and will keep it in mind for future studies of new standardized batches of PPAb that we plan to obtain from our A. bogotensis crops grown under good agricultural practices.

  1. Pharmacokinetic studies and tissue distribution of PPAb, antiviral effect evaluation performed in animals may be better than simple in vitro toxicity on human intestinal cells.

Answer: We agree with the reviewer. Fortunately, we have identified the main compounds of the active fraction to perform in vivo studies. Moreover, we previously did an acute oral toxicity test with the first batch of the PPAb (limit test) in pigs (1000 and 5000 mg/kg body weight x 14 days / OECD Guidelines for the testing of chemicals - 409, 401 and 423). The results showed the absence of observable signs compatible with intoxication produced by the evaluated PPAb or by the vehicle during the 14 days of administration. Our patent (reference 23) includes these data.

Minor concerns:

  1. Line 231 “conducted in triplicate were analised and statistical significance (p < 0.05) was determined by “analised” should be “analyzed”.

Answer: British English spelling was changed to American English spelling.

  1. For the figure legends, different graphs were marked as capitalized A/B, while in the results part, they are written in lower-case letters(a/b)

Answer: All figure legends, graphs, and paragraphs in the manuscript were modified according to the reviewer's comment.

Finally, the following revisions were also made:

- The abbreviation PTP (phytotherapeutic prototype) was replaced by PPAb (phytotherapeutic prototype of A. bogotensis) in all graphs.

- Figures 4A/4B and 6A/6B are now presented together as Figure 4 and Figure 6.

- In the Discussion section (Lines 379 -386), an analysis of the differences in percentages of inhibition of viral infection between batches 1 and 2 of the PPAb was added.

- The writing in English was revised again throughout the document. 

Round 2

Reviewer 1 Report

None.